# Neural Optimal Transport with Lagrangian Costs

**Aram-Alexandre Pooladian**[1,3]    **Carles Domingo-Enrich**[2,3]    **Ricky Tian Qi Chen**[3]    **Brandon Amos**[3]

[1]Center for Data Science, New York University
[2]Courant Institute of Mathematical Sciences, New York University
[3]FAIR, Meta

## Abstract

We investigate the optimal transport problem between probability measures when the underlying cost function is understood to satisfy a *least action principle*, also known as a *Lagrangian* cost. These generalizations are useful when connecting observations from a physical system where the transport dynamics are influenced by the geometry of the system, such as obstacles (*e.g.*, incorporating barrier functions in the Lagrangian), and allows practitioners to incorporate *a priori* knowledge of the underlying system such as non-Euclidean geometries (*e.g.*, paths must be circular). Our contributions are of computational interest, where we demonstrate the ability to efficiently compute geodesics and amortize spline-based paths, which has not been done before, even in low dimensional problems. Unlike prior work, we also output the resulting *Lagrangian optimal transport map* without requiring an ODE solver. We demonstrate the effectiveness of our formulation on low-dimensional examples taken from prior work. The source code to reproduce our experiments is available at https://github.com/facebookresearch/lagrangian-ot.

## 1 INTRODUCTION

Computational efforts in optimal transport traditionally revolve around the squared-Euclidean cost $\frac{1}{2}\|x - y\|^2$. This cost has a connection to convex functions via Brenier's theorem [Brenier, 1991], and has allowed for both numerical analysts [Jacobs and Léger, 2020] and machine learning researchers [Bunne et al., 2022b, Amos, 2023, Korotin et al., 2019] to push the boundaries of computational optimal transport in recent years. This connection has also been influential in domains such as economics and statistics [Car-

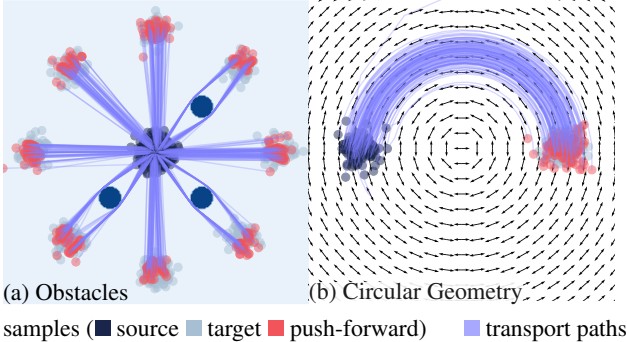

(a) Obstacles    (b) Circular Geometry

samples (■ source ■ target ■ push-forward)    ■ transport paths

Figure 1: Optimal transport paths with Lagrangian costs on the obstacles setting from Liu et al. [2022] and circular geometry from Scarvelis and Solomon [2023].

lier et al., 2016, Chernozhukov et al., 2017], high-energy particle physics [Manole et al., 2022], computational biology [Schiebinger et al., 2019, Bunne et al., 2021, 2022a], computer vision [Feydy et al., 2017], among others.

However, there is little reason practitioners should *default* to this cost in their applications, where often they know that paths will not be straight lines, or have obstacles that must be avoided. The purpose of this paper is to provide a computational framework that allows practitioners to enforce transport with more general costs that can incorporate such geometries. To this end, our goal is to numerically solve the optimal transport problem when the underlying cost of displacement is governed by a *least action principle*. For two points $x, y \in \mathbb{R}^d$, the *displacement cost* $c(x, y)$ is

$$c(x, y) = \inf_{\gamma \in \mathcal{C}(x,y)} \left\{ \int_0^1 \mathcal{L}(\gamma_t, \dot{\gamma}_t) \, \mathrm{d}t \right\}, \qquad (1)$$

where $\mathcal{C}(x, y)$ is the set of smooth, time dependent curves $\gamma$ that connect $x$ and $y$ such that $\gamma_0 = x$ and $\gamma_1 = y$, and $\mathcal{L} : \mathbb{R}^d \times \mathbb{R}^d \to \mathbb{R} \cup \{+\infty\}$ is the *Lagrangian function* which ultimately governs the cost of transport. The Lagrangian takes as arguments the position of the curve $\gamma_t \in \mathbb{R}^d$ at time $t$ and the velocity at that point $\dot{\gamma}_t := \frac{\mathrm{d}\gamma_t}{\mathrm{d}t}$. This definition is

inspired by Lagrangian mechanics: equations of motion that are based on energies of a system, rather than forces. As outlined in Villani [2009, Chapter 7], and briefly discussed in section 2.2, this notion of cost can be lifted to the space of probability measures, instead of just being between two fixed points over a space.

If $\mathcal{L}(x, v) = \frac{1}{2}\|v\|^2$, then $c(x, y)$ recovers the squared-Euclidean distance of transport (*cf.* Benamou and Brenier [2000]). However, the Lagrangians that we consider are more general. They not only impact the transport destination, but also the optimal path (instead of just straight lines). Examples include

1) *potential energy terms* (see example 2), with

$$\mathcal{L}(x, v) = \frac{1}{2}\|v\|^2 - U(x),$$

2) *position-dependent costs* (see example 3), with

$$\mathcal{L}(x, v) = \frac{1}{2}\|v\|^2_{A(x)} := \frac{1}{2}v^\top A(x)v,$$

where $U : \mathbb{R}^d \to \mathbb{R}$ is a potential function, and $A : \mathbb{R}^d \to \mathbb{S}^d_{++}$ is a (positive definite) matrix-valued function.

Figure 1 contains two Lagrangian optimal transport problems. In fig. 1(a), smooth potential functions act as obstacles between the source Gaussian and the 8-Gaussian mixture. In fig. 1(b), the cost of displacement is lowest along circular trajectories. We stress that, despite being non-standard notions of cost, there still exists an optimal transport map (see eq. (6)) expressed explicitly as the minimizer to an optimization problem, and the paths are unique. These notions of cost have appeared in a variety of works (see e.g., Koshizuka and Sato [2022], Scarvelis and Solomon [2023], Liu et al. [2022]), though none of these approaches provide *deterministic* mappings (i.e., source-to-target maps in one function evaluation), nor do they provide optimal paths.

**Main contributions.** We aim to fill this gap in the literature on computational optimal transport, where the cost function follows eq. (1) with potential energies or position-dependent costs, and the measures are in an underlying continuous space. Our two main goals, which go hand-in-hand, are to:

(1) Compute the Lagrangian optimal transport maps,

(2) Compute the resulting *paths* for these maps.

We want to emphasize that, to the best of our knowledge, these approaches have not been considered in the machine learning literature related to *unregularized* optimal transport. Indeed, since the cost function eq. (1) is itself a minimization problem, the resulting OT problem (see section 2) is a *bilevel* optimization problem, wherein lies the difficulty of these general costs. Our work acts as a first step to tackling these optimization problems in a principled manner.

Borrowing inspiration from the existing literature, we consider two categories of optimal transport problems: i)

transport between a *pair* of probability measures $(\mu, \nu)$, and ii) transport between *consecutive pairs* of measures $\{(\rho_i, \rho_{i+1})\}_{i=0}^{K-1}$. In i), we assume the practitioner is interested in modeling physical systems, and has access to their Lagrangian of interest, either through the potential function $U$ or position-dependent metric $A$, and wants to know the optimal displacement and cost between $\mu$ and $\nu$. For ii), the practitioner has access to samples from $K$ probability measures, which they believe to be traversing optimally under some underlying Riemannian metric which is not known. We *learn* this Riemannian metric to uncover the geometry of the space. We stress that both modifications allow for the practitioner to employ in-domain knowledge to the cost function, as opposed to the squared-Euclidean cost, which remains information agnostic.

Our approach involves parameterizing the Lagrangian optimal transport maps and paths using neural networks. The non-standard cost leads to two computational challenges for obtaining 1) the displacement cost eq. (1) and minimizing path, 2) the $c$-transform of the Lagrangian cost. We overcome both of these by using amortized optimization (*e.g.* as in Amos et al. [2023]) to obtain approximate solutions. In the two tasks we consider (Lagrangian optimal transport between two measures, and Riemannian metric learning through a sequence of pairs of measures), we outperform existing baselines on data taken from the respective papers.

## 2 BACKGROUND ON OPTIMAL TRANSPORT

### 2.1 KANTOROVICH PRIMAL-DUAL PROBLEMS AND OPTIMAL TRANSPORT MAPPINGS

Optimal transport can be written as several equivalent infinite-dimensional optimization problems, which we outline below under mild conditions. We refer the interested reader to Santambrogio [2015] or Villani [2009] for a more detailed discussion. Let $\mu \in \mathcal{P}(\mathcal{X})$ and $\nu \in \mathcal{P}(\mathcal{Y})$ be two probability measures defined on $\mathcal{X}$ and $\mathcal{Y}$, respectively, which are complete, separable metric spaces (for simplicity, one can consider $\mathbb{R}^d$ endowed with the Euclidean metric). Let $c : \mathcal{X} \times \mathcal{Y} \to \mathbb{R}$ be a lower semicontinuous, real-valued cost function (for simplicity, one can consider any bounded convex cost function).

The *primal (Kantorovich) formulation*, attributed to Kantorovitch [1942], is given by

$$\mathrm{OT}_c(\mu, \nu) := \inf_{\pi \in \Gamma(\mu, \nu)} \iint_{\mathcal{X} \times \mathcal{Y}} c(x, y)\, \mathrm{d}\pi(x, y), \quad (2)$$

where $\Gamma(\mu, \nu) \subset \mathcal{P}(\mathcal{X} \times \mathcal{Y})$ is the set of transportation couplings between $\mu$ and $\nu$ *i.e.*, $\pi \in \Gamma(\mu, \nu)$ if

$$\int_{\mathcal{Y}} \mathrm{d}\pi(x, y) = \mathrm{d}\mu(x), \quad \int_{\mathcal{X}} \mathrm{d}\pi(x, y) = \mathrm{d}\nu(x). \quad (3)$$

Under our specifications on the cost function, an equivalent optimization problem called the *dual (Kantorovich) formulation*, cf. Villani [2009, Theorem 5.10], is

$$\mathrm{OT}_c(\mu, \nu) = \sup_{g \in L^1(\nu)} \int g^c(x)\, d\mu(x) + \int g(y)\, d\nu(y)\,, \quad (4)$$

where $L^1(\nu)$ is the set of integrable functions with respect to $\nu$, and $g^c$ is the *c-transform of g*, written

$$g^c(x) := \inf_{y \in \mathcal{Y}} J(y; x) \text{ where } J(y; x) := c(x, y) - g(y)\,. \quad (5)$$

When attained, the minimizer of eq. (5) is

$$\hat{y}(x; c, g) := \operatorname*{argmin}_{y \in \mathcal{Y}} \{c(x, y) - g(y)\}\,. \quad (6)$$

When the supremum in eq. (4) is attained, we write $\hat{g}$ as the maximizer, called the optimal Kantorovich potential. We define the *optimal transport map* associated to the cost $c$ as the minimizer $\hat{y}(\cdot; c, \hat{g})$, which is eq. (6) applied to the optimal Kantorovich potential. Given $x \in \mathcal{X}$, $\hat{y}(x; c, \hat{g})$ corresponds to the optimal displacement from $\mu$ to $\nu$.

## 2.2   LAGRANGIAN OPTIMAL TRANSPORT (LOT)

We now suppose our probability measures exist on compact subsets $\mathcal{X} = \mathcal{Y} \subseteq \mathbb{R}^d$. We associate the cost of displacing $x$ to $y$ with an *action* that is to be minimized over a time horizon $[0, 1]$. Borrowing terminology from physics, these actions will take the form of *Lagrangian* functionals, which are functions that depend on the position of a curve $\gamma_t$, its velocity, $\dot{\gamma}_t$, and time $t \in [0, 1]$;

$$(\gamma_t, \dot{\gamma}_t) \mapsto \mathcal{L}(\gamma_t, \dot{\gamma}_t)\,, \quad (7)$$

where curves in $\mathcal{C}$ are understood to be smooth and absolutely continuous curves over $\mathbb{R}^d$, indexed by time in $[0, 1]$, cf. Villani [2009, Chapter 7]. The Lagrangian induces an *action* or *energy* $E$ on curves defined by

$$E(\gamma; x, y) = \left\{ \int_0^1 \mathcal{L}(\gamma_t, \dot{\gamma}_t)\, dt \right\}\,. \quad (8)$$

The *cost of displacement* is then given by

$$c(x, y) = \inf_{\gamma \in \mathcal{C}(x, y)} E(\gamma; x, y)\,. \quad (9)$$

Though initially defined between two points on the manifold, this cost can be appropriated "lifted" to the space of probability measures, resulting in what is known as *Lagrangian Optimal Transport* (LOT). Indeed, under mild assumptions on $\mathcal{L}$, the generalized notion of transport vis-à-vis minimizers to eq. (5) is defined. A thorough discussion is found in Villani [2009, Chapter 7], specifically Theorem 7.21 and Remark 7.25. The following conditions are sufficient for eq. (9) to define a valid notion of transport: $\mathcal{L}$ is

twice continuously differentiable and strictly convex in $v$, with $\nabla_v^2 \mathcal{L} \succ 0$ everywhere, and $\mathcal{L}$ does not depend (explicitly) on $t$. These conditions are satisfied in *all* our problem considerations. Thus, when $c$ is a cost of the form eq. (9), we refer to the Lagrangian optimal transport map (or LOT map) as the minimizer to eq. (6) under this cost.

*Remark* 1. For simplicity, we present the background for manifolds $(\mathbb{R}^d, g)$ where $g$ is potentially a non-Euclidean metric. These same discussions hold when we instead consider a general smooth Riemannian manifold $\mathcal{M}$ and its associated metric $g$; cf. e.g., Feldman and McCann [2002].

**Example 1** (Euclidean distances, cf. Benamou and Brenier [2000]). *The squared Euclidean distance is recovered, i.e., $c(x, y) = \|x - y\|_2^2$, by taking the Lagrangian as the kinetic energy:*

$$\mathcal{L}(\gamma_t, \dot{\gamma}_t, t) = \tfrac{1}{2} \|\dot{\gamma}_t\|^2\,. \quad (10)$$

*Indeed, $\mathcal{L}$ is twice differentiable with $\nabla_v^2 \mathcal{L} = I \succ 0$, which satisfies our conditions.*

**Example 2** (Obstacles and other potential functions). *One can add a potential function (not to be confused with Kantorovich potentials from section 2.1) $U : \mathbb{R}^d \to \mathbb{R}$, to the kinetic energy, resulting in the Lagrangian*

$$\mathcal{L}(\gamma_t, \dot{\gamma}_t) = \frac{1}{2} \|\dot{\gamma}_t\|^2 - U(\gamma_t)\,. \quad (11)$$

*Again, $\nabla_v^2 \mathcal{L} = I \succ 0$. We require $U$ to be sufficiently smooth in order for $\mathcal{L}$ to be twice continuously differentiable. The function $U$ provides a way of specifying how "easy" or "hard" it is to pass through regions of the space. This includes the obstacles as in fig. 1(a) and fig. 2 where the potential takes low values and prevents the paths from crossing them.*

**Example 3** (Squared geodesic distances on Riemannian manifolds). *Example 1 can be extended to non-Euclidean manifolds. In $\mathbb{R}^d$, the metric at a point $x \in \mathbb{R}^d$ is given by the inner product $\langle u, v \rangle_x := \langle u, A(x)v \rangle$ for any $u, v \in \mathbb{R}^d$, for $A(\cdot) : \mathbb{R}^d \to \mathbb{S}_{++}^d$ positive-definite, giving the Lagrangian*

$$\mathcal{L}(\gamma_t, \dot{\gamma}_t; A) = \frac{1}{2} \|\dot{\gamma}_t\|_{A(\gamma_t)}^2\,. \quad (12)$$

*Here, $\nabla_v^2 \mathcal{L} = A(\gamma_t)$, so we require $A(\cdot) \succ 0$ to satisfy the criteria of Theorem 7.21 and Remark 7.25 from Villani [2009].*

Example 3 shows the circular geometry in fig. 2(a) where the metric is given by the positive-definite matrix

$$A(x) = \begin{pmatrix} \frac{x_1^2}{\|x\|^2} & 1 - \frac{x_1 x_2}{\|x\|^2} \\ 1 - \frac{x_1 x_2}{\|x\|^2} & \frac{x_2^2}{\|x\|^2} \end{pmatrix}\,. \quad (13)$$

**Algorithm 1** Neural Lagrangian Optimal Transport (NLOT)

---

**inputs:** measures $\mu$ and $\nu$, Kantorovich potential $g_\theta$, $c$-transform predictor $y_\zeta$, and spline predictor $\varphi_\eta$
**while** unconverged **do**
    sample batches $\{x_i\}_{i=1}^N \sim \mu$ and $\{y_i\}_{i=1}^N \sim \nu$
    obtain the amortized $c$-transform predictor $y_\zeta(x_i)$ for $i \in [N]$
    fine-tune the $c$-transform by numerically solving eq. (6), warm-starting with $y_\zeta(x_i)$
    update the potential with gradient estimate of $\nabla_\theta \ell_{\text{dual}}$ (eq. (15))
    update the $c$-transform predictor $y_\zeta$ using a gradient estimate of eq. (17)
    update the spline predictor $\varphi_\eta$ using a gradient estimate of eq. (19)
**end while**
**return** optimal parameters $\theta, \phi, \eta$

---

## 3 LAGRANGIAN OT BETWEEN TWO MEASURES VIA NEURAL NETWORKS

We first focus on computationally solving for the Kantorovich dual in eq. (4) between two measures $\mu \in \mathcal{P}(\mathcal{X})$ and $\nu \in \mathcal{P}(\mathcal{Y})$ when the cost function is of the form eq. (11) or eq. (12). All components of the Lagrangian are known, *i.e.*, the Lagrangian potential $U$ or the underlying metric $A(\cdot)$ is known, and we assume access to samples from $\mu$ and $\nu$. The Kantorovich potential $g \in L^1(\nu)$ in eq. (4) is a function $g : \mathcal{Y} \to \mathbb{R}$. We present a detailed explanation below; Alg. 1 summarizes our solution.

We follow recent neural optimal transport methods, *e.g.*, Taghvaei and Jalali [2019], Makkuva et al. [2020], Korotin et al. [2019], Fan et al. [2021a], Amos [2023], and represent the Kantorovich potential as a neural network $g_\theta$ with parameters $\theta$. With this parameterization, we recast eq. (4) as $\max_\theta \ell_{\text{dual}}(\theta)$ where

$$\ell_{\text{dual}}(\theta) := \int g_\theta^c(x) \, \mathrm{d}\mu(x) + \int g_\theta(y) \, \mathrm{d}\nu(y) \qquad (14)$$

and the $c$-transform $g_\theta^c$ incorporates the Lagrangian function. We optimize eq. (14) via gradient descent, where the derivative in the parameters is

$$\nabla_\theta \ell_{\text{dual}}(\theta) = \int \nabla_\theta g_\theta^c(x) \, \mathrm{d}\mu(x) + \int \nabla_\theta g_\theta(y) \, \mathrm{d}\nu(y), \quad (15)$$

and $g_\theta^c$ is differentiated with *Danskin's envelope theorem* [Danskin, 1966, Bertsekas, 1971], *i.e.*,

$$\nabla_\theta g_\theta^c(x) = \nabla_\theta J(\hat{y}(x); x, c, g_\theta) = -\nabla_\theta g_\theta(\hat{y}(x)). \quad (16)$$

We follow Taghvaei and Jalali [2019], as well as other OT work based on neural networks, and approximate eq. (14) and eq. (15) with Monte-Carlo estimates of the integrals as they are not computable in closed-form. Computing these estimates still requires overcoming the following challenges:

**Challenge 1** (Computing the $c$-transform). *Estimating eq. (14) and eq. (15) require obtaining the $c$-transform $g_\theta^c$ and the corresponding minimizing point $\hat{y}(x; c, g)$. This requires solving the optimization problem in eq. (5) for **every** $x$, which does not have a closed-form solution.*

Prior OT work for the squared-Euclidean cost settings had to overcome a similar challenge when the $c$-transform becomes the Fenchel or convex conjugate operation: Taghvaei and Jalali [2019], Korotin et al. [2021] use numerical solvers such as L-BFGS, Adam, and other gradient-based methods, Makkuva et al. [2020], Korotin et al. [2019, 2021] use an amortized approximation, and Amos [2023] combines the amortized approximation with a numerical solver. For $c$-transforms, Fan et al. [2021a] uses an amortized approximation to overcome challenge 1.

We follow these works and overcome challenge 1 by amortizing the solution to eq. (5). This involves parameterizing an approximate $c$-transform map $\hat{y}_\zeta \approx \hat{y}$ that we learn with a regression-based loss

$$\min_\phi \int \|\hat{y}(x) - y_\zeta(x)\| \, \mathrm{d}\mu(x). \qquad (17)$$

The conjugation model $\hat{y}_\zeta$ is only an approximation and may be inaccurate as the potential $g_\theta$ changes during training. An inaccurate approximation to the $c$-transform results in a poor approximation to the objective in eq. (14); to improve it, we follow Amos [2023] and fine-tune the $c$-transform prediction with a few steps of L-BFGS to solve eq. (6), and warm-start it with the amortized prediction.

**Challenge 2** (Computing the cost $c$). *Evaluating the Lagrangian cost $c$ that arises in the $c$-transform in eqs. (5) and (17) involves solving the optimization problem in eq. (9) over paths. While closed-form solutions exist for simple manifolds, e.g., straight paths on Euclidean space or great arcs on spherical manifolds, the more general settings we consider do not admit closed-form solutions and need to be numerically solved.*

Computationally representing paths and solving for Riemannian geodesics and Lagrangian paths in eq. (9) outside of the context of optimal transport is an active research area. We follow Beik-Mohammadi et al. [2021], Detlefsen et al. [2021] and parameterize the space of paths between $x$ and $y$ with a cubic spline $\gamma_\varphi(x, y)$ (where the parameters are $\varphi$). This spline parameterization transforms the optimization problem in eq. (9) to an optimization problem over the

continuous-valued parameters of the spline as

$$\varphi^{\star}(x,y) := \underset{\varphi \in \Phi(x,y)}{\operatorname{argmin}} E(\varphi; x, y) \qquad (18)$$

where $E(\varphi; x, y) := \left\{ \int_0^1 \mathcal{L}((\gamma_\varphi)_t, (\dot{\gamma}_\varphi)_t)\, \mathrm{d}t \right\}$ and $\Phi(x,y)$ is the space of cubic splines between $x$ and $y$; see appendix A for more details.

Solving eq. (18) for every $c$-transform within every evaluation for the OT cost is computationally intractable, so we propose to amortize the path computation using objective-based amortization as in Amos et al. [2023]. Given points $x$ and $y$, we parameterize the spline amortization model with $\varphi_\eta(x,y) \approx \varphi^{\star}(x,y)$, where $\eta$ represents the weights of a neural network. We train $\varphi_\eta$ to compute the paths necessary for the Lagrangian cost, *i.e.*,

$$\min_\eta \int E(\varphi_\eta; x, \hat{y}(x))\, \mathrm{d}\mu(x)\,. \qquad (19)$$

## 3.1 EXPERIMENTS

We consider Lagrangians of the form

$$L(x, v) = \tfrac{1}{2}\|v\|^2 - U(x)\,, \qquad (20)$$

with $U : \mathbb{R}^d \to \mathbb{R}$, where $U$ can take the form of a barrier that distorts the transport path from being a straight line. For modeling hard constraints (recall the three circular obstacles in fig. 1(a)), we model $U$ as a smooth (but sharp) barrier function, which preserves smoothness in $L$. Precise definitions of the functions are deferred to appendix B. In fig. 2, we learn the optimal transport map between two measures with a box or slit constraint, or a hill or a well. In fig. 1(a), we learn the optimal transport map between a Gaussian and a Gaussian mixture, with barriers (three circular barriers). A single training run for all of the experimental settings takes approximately 1-3 hours on our NVIDIA Tesla V100 GPU.

Our examples in fig. 2 are taken from Koshizuka and Sato [2022], and is our main benchmark for this task. Their approach is based on learning Schrodinger Bridges with neural networks, called NLSB (Neural Lagrangian Schrodinger Bridge). Therein, the authors use stochastic differential equations (SDEs) to model the system, and, using a numerical solver, integrate in order to obtain a path, and thus a mapping. In contrast, we *directly* learn the mappings and paths simultaneously. Table 1 compares the marginal distribution of the pushforwards from with fresh samples, and fig. 2 contains the trajectories. In all scenarios, our mapping gives a more faithful estimate of the target distribution.

We want to stress that, even though we present paths, our Lagrangian OT maps are also learned and deterministic, whereas the NLSB method requires integrating and SDE to obtain the final displacement. Finally, NLSB presents two path variants: stochastic paths, and expected paths. The

Table 1: Marginal 2-Wasserstein errors (scaled by 100x) of the push-forward measure on the synthetic settings from Koshizuka and Sato [2022].

|  | box | slit | hill | well |
|---|---|---|---|---|
| NLOT (ours) | **1.6 ± 0.2** | **1.3 ± 0.2** | **1.8 ± 1.3** | **1.3 ± 0.3** |
| NLSB (stochastic) | 2.4 ± 0.6 | **1.3 ± 0.4** | 2.0 ± 0.1 | 2.6 ± 1.6 |
| NLSB (expected) | 6.0 ± 0.5 | 17.6 ± 1.8 | 4.0 ± 0.9 | 16.1 ± 3.5 |

*Results are from training three trials for every method.

former is the trajectory of a bona fide SDE, and the latter is the average trajectory starting from a given point. On one hand, the expected path follows an ODE, and is easier to integrate; on the other, the trajectories are more degenerate, and are significantly less meaningful.

## 3.2 RELATED WORK

**Lagrangian Schrödinger bridges.** Koshizuka and Sato [2022] and Liu et al. [2022, 2024] are the closest for this subproblem that we consider in section 3. The former studies the *Stochastic Optimal Transport* (SOT) problem, which amounts to optimal transport on path space, with the path dictated by a stochastic differential equation (SDE). The authors consider Lagrangian costs and use neural SDEs to model the trajectories. Liu et al. [2022, 2024] investigate the generalized *Schrödinger Bridge Problem* (SBP), which can be distilled to optimal transport with entropic regularization [Léonard, 2012], also using neural SDEs, and have a particular focus on mean-field games, which is not a focus of this work (see for example Lin et al. [2020], Ruthotto et al. [2020], Ding et al. [2022]) The SBP is also a special case of SOT; see Koshizuka and Sato [2022, Figure 2].

**Estimation and applications of optimal transport maps under the squared-Euclidean cost.** Apart from machine learning communities, the squared-Euclidean cost has also garnered much interest in traditional domains. For example, statistical estimation of optimal transport maps for the squared-Euclidean cost started with Hütter and Rigollet [2021], followed swiftly by Deb et al. [2021], Manole et al. [2021], Pooladian and Niles-Weed [2021], to name a few. Applications of optimal transport in machine learning often revolve around "generative modeling", where the notion of a learned transport map allows us to generate new samples from a target measure from which we only have access to samples (*e.g.*, generating a new image). Examples of such works include Huang et al. [2021], Finlay et al. [2020a,b], Lipman et al. [2022], Pooladian et al. [2023], Onken et al. [2021], Rout et al. [2021], Bousquet et al. [2017], Balaji et al. [2020], Seguy et al. [2018], Tong et al. [2023]. The work of Schiebinger et al. [2019] had a cascading effect in the machine learning community, popularizing the ability to predict single-cell genome expressions through optimal transport using limited data. Examples include Bunne et al.

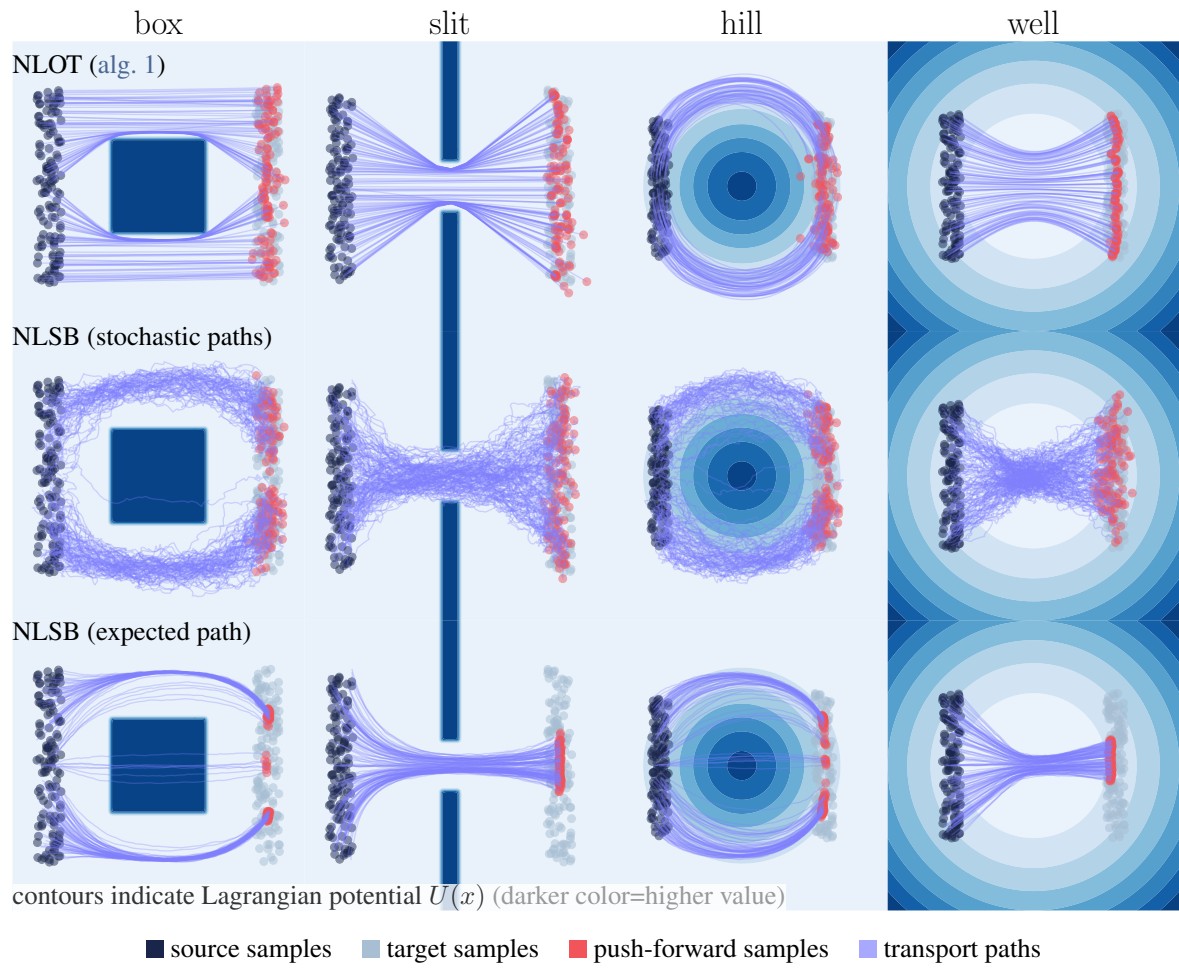

|  | box | slit | hill | well |
|---|---|---|---|---|
| NLOT (alg. 1) | | | | |
| NLSB (stochastic paths) | | | | |
| NLSB (expected path) | | | | |

contours indicate Lagrangian potential $U(x)$ (darker color=higher value)

■ source samples ■ target samples ■ push-forward samples ▮ transport paths

Figure 2: Paths on the Neural Lagrangian Schrödinger Bridge (NLSB) datasets [Koshizuka and Sato, 2022].

[2021, 2022b,a], Lübeck et al. [2022] and Tong et al. [2020].

**Estimation of optimal transport maps for other notions of cost.** One can generalize the notion of a Brenier-map in closed form by considering a specific family of cost functions. We call a convex cost function *translation invariant* if $c(x, y) \coloneqq h(x - y)$ with $h : \mathbb{R}^d \to \mathbb{R}$ convex. To the best of our knowledge, the pursuit of estimating such maps has been seldom, apart from Fan et al. [2021b] and the recent works Cuturi et al. [2023], Uscidda and Cuturi [2023], Klein et al. [2023]. Other applications include defining notions of optimal transport between datasets or different spaces [Nekrashevich et al., 2023, Alvarez-Melis and Fusi, 2020, Alvarez-Melis and Jaakkola, 2018].

## 4 METRIC LEARNING WITH NLOT

The following set of experiments is inspired from recent works such as Tong et al. [2020], Bunne et al. [2022b], Zhang et al. [2022], Schiebinger et al. [2019] that assume data is obtained as sparse pairs of sequences $\{\rho_i, \rho_{i+1}\}_{i=1}^{K-1}$ (such a setup arises in single-cell genomic profiling, for ex-

ample). At the core of these methods is the crucial assumption that the space is Euclidean, which allows the authors to leverage various facts about optimal transport maps arising from convex costs. In contrast, Scarvelis and Solomon [2023] considers the perspective that the data in fact arises from a non-Euclidean Riemannian manifold, where the underlying metric is given by some twisted inner product with respect to a positive definite matrix (recall example 3 in section 2.2).

With this perspective in mind, we now consider optimal transport problems where the ground-truth displacement is given by geodesics induced by non-Euclidean geometries, like eq. (13). However, we crucially *do not* assume knowledge of the underlying positive-definite matrix-valued function $A(\cdot)$ that induces the Riemannian geometry. Our goal is to instead *learn* $A(\cdot)$ on the basis of sequential pairs of probability measures $\{(\rho_i, \rho_{i+1})\}_{i=1}^{K-1}$, as well as the final transportation mappings and paths.

Let $A_\vartheta$ be the neural network parameterization of a positive-definite matrix, with the network weights given by $\vartheta$. The matrix-valued function $A_\vartheta$ then induces the cost $c_\vartheta$

---

**Algorithm 2** Metric learning with NLOT

---

**inputs:** measures $\{(\rho_i, \rho_{i+1})\}_{i=1}^{K-1}$, metric $A_\vartheta$, potentials $g_{\theta_i}$, $c$-transform predictors $y_{\phi_i}$, spline predictors $\varphi_{\eta_i}$,
**while** unconverged **do**
    update $\vartheta$ using $\nabla_\vartheta \ell_{\text{dual}}$ (with the terms in eq. (25))
    update the OT approximation $\theta_i, \phi_i, \eta_i$ with an iteration of alg. 1
**end while**
**return** optimal parameters $\vartheta, \theta_i, \phi_i, \eta_i$

---

$$c_\vartheta(x,y) := \inf_{\gamma \in \mathcal{C}(x,y)} \left\{ \int_0^1 \tfrac{1}{2} \|\dot\gamma_t\|_{A_\vartheta(\gamma_t)}^2 \, dt \right\}. \quad (21)$$

Following Scarvelis and Solomon [2023], our goal is to learn a metric that results in a geometry with a minimal OT cost, *i.e.*, $\min_\vartheta \ell_{\text{metric}}(\vartheta)$ where

$$\ell_{\text{metric}}(\vartheta) := \frac{1}{K} \sum_{i=1}^{K-1} \text{OT}_{c_\vartheta}(\rho_i, \rho_{i+1}). \quad (22)$$

We use the neural networks from section 3 to approximate the OT maps, resulting in

$$\ell_{\text{metric}}(\vartheta) \approx \max_{\{\theta_i\}_{i=1}^{K-1}} \frac{1}{K} \sum_{i=1}^{K-1} \ell_{\text{dual}}(\theta_i; \rho_i, \rho_{i+1}, \vartheta). \quad (23)$$

Altogether, we aim to solve the following min-max optimization problem

$$\min_\vartheta \max_{\{\theta_i\}_{i=1}^{K-1}} \frac{1}{K} \sum_{i=1}^{K-1} \ell_{\text{dual}}(\theta_i; \rho_i, \rho_{i+1}, \vartheta) \quad (24)$$

with alternating descent-ascent. For a fixed metric $A_\vartheta$, the inner maximization problem is the same as section 3, but with $K-1$ networks. The only difference is the outer minimization step, which we compute efficiently via sequential applications of the envelope theorem. Noting that only the first term in eq. (22) depends on $A_\vartheta$, the gradient of eq. (23) is given by

$$\begin{aligned}
\nabla_\vartheta \ell_{\text{dual}}(\theta; \rho_i, \rho_{i+1}, \vartheta) &= \nabla_\vartheta \int g^{c_\vartheta} \, d\rho_i \\
&= \int \nabla_\vartheta g^{c_\vartheta} \, d\rho_i \\
&= \int \nabla_\vartheta c_\vartheta(x, \hat{y}(x)) \, d\rho(x) \\
&= \int \nabla_\vartheta E_\vartheta(\varphi_{\eta_i}, x, \hat{y}(x))) \, d\rho_i .
\end{aligned} \quad (25)$$

The full update of $A_\vartheta$ then takes the average gradient of these $K-1$ gradient computations. Thus, the inner maximization step requires $K-1$ applications of alg. 1, and the outer minimization step freezes the inner parameters, leaving only an average update for $A_\vartheta$. We stress that a primary difference in this setting limited finite-sample access to the $K-1$ measures from which we are to learn the ground-truth metric, and output paths and optimal transport maps. Alg. 2 overviews the general algorithm.

## 4.1 EXPERIMENTS

We consider three ground-truth Riemannian metrics $A(\cdot)$, which are given by the arrows in fig. 4 and fig. 3. To be precise, the grey arrows in the figures show the direction of the smallest eigenvector at that point *i.e.*, the easiest direction to move in. Note that fig. 3(a) is the circle metric from eq. (13), and the other two metrics are non-smooth metrics that cause splitting fig. 3(b) or reflections fig. 3(c). In each task, we are given samples from $K-1$ pairs of probability measures which were generated according to these Riemannian metrics. Our task is to learn the metric on the basis of the samples alone, and ideally recover the transport path exactly. The precise formulas for $A(\cdot)$ and descriptions of the learning tasks are in appendix C.

We parameterize the metric $A_\vartheta$ to predict a rotation matrix $R_\vartheta(x)$ of a fixed matrix $B$, *i.e.*,

$$A_\vartheta(x) := R_\vartheta(x) B R_\vartheta(x)^\top, \quad B := \begin{bmatrix} 1 & 0 \\ 0 & 0.1 \end{bmatrix}, \quad (26)$$

and the rotation matrix

$$R_\vartheta(x) := \begin{bmatrix} \cos\theta_\vartheta(x) & -\sin\theta_\vartheta(x) \\ \sin\theta_\vartheta(x) & \cos\theta_\vartheta(x) \end{bmatrix} \quad (27)$$

is obtained from predicting a rotation $\theta_\vartheta(x)$ from the input $x$. While this seemingly ad hoc approach appears limited to rotations, we find this to be far from the truth, as this method succeeds in learning three different geometries. The eigenvalues of $A_\vartheta(x)$ are always the eigenvalues of $B$ (1 and 0.1), and parameterizing the rotation forces $B$ to be rotated so the data movement is along the smallest eigenvector as in fig. 3.

We quantify our ability to recover the ground-truth metric through the alignment score from Scarvelis and Solomon [2023]:

$$\ell_{\text{align}}(A, \hat{A}) := \frac{1}{d|\mathcal{D}|} \sum_{x \in \mathcal{D}} \sum_{i=1}^{d} |u_i(x)^\top \hat{u}_i(x)|, \quad (28)$$

where $\mathcal{D}$ is a finite discretization of the space, and $u_i(x)$ (resp. $\hat{u}_i(x)$) is the (unit) eigenvector with eigenvalue $\lambda_i$ (resp. $\hat{\lambda}_i$) for the matrix $A(x)$ (resp. $\hat{A}(x)$). Our results are reported in table 2, where we perform the same experiment over three randomized trials, and report the same metric

Table 2: Alignment scores $\ell_{\text{align}} \in [0, 1]$ for metric recovery in fig. 4. (higher is better)

| | Circle | Mass Splitting | X Paths |
|---|---|---|---|
| Metric learning with NLOT (ours) | **0.997 ± 0.002** | **0.986 ± 0.001** | **0.957 ± 0.001** |
| Scarvelis and Solomon [2023] | 0.995 | 0.839 | 0.916 |

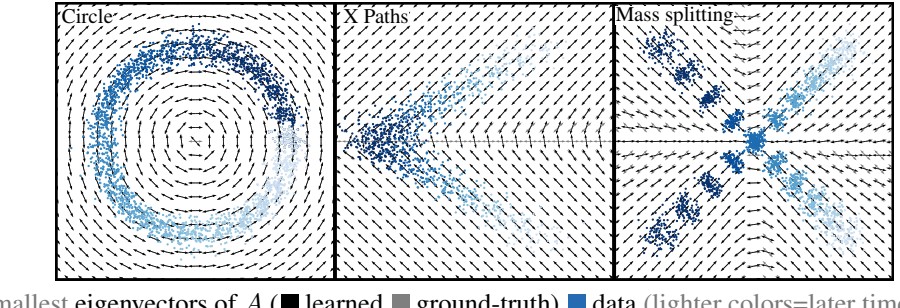

smallest eigenvectors of $A$ (■ learned ■ ground-truth) ■ data (lighter colors=later time)

Figure 3: We successfully recover the metrics on the settings from Scarvelis and Solomon [2023].

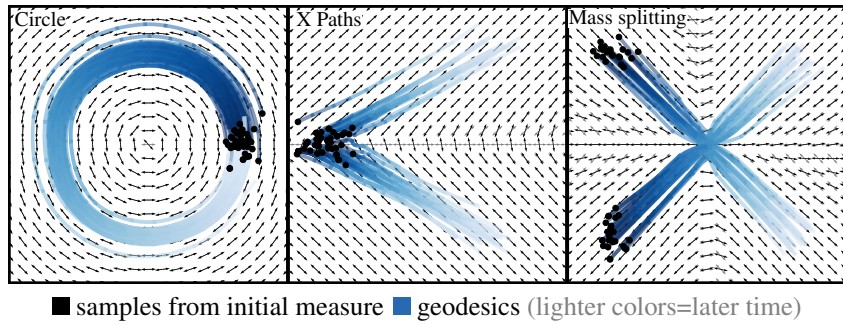

■ samples from initial measure ■ geodesics (lighter colors=later time)

Figure 4: Our transport geodesics are able to reconstruct continuous versions of the original data that can predict the movement of individual particles given only samples from the first measure.

values from Scarvelis and Solomon [2023]. Notably, we see a roughly 17% improvement in the "Mass Splitting" example, with near-perfect recovery. Finally, in fig. 4, we plot our fitted geodesics that are learned from the data. Unlike Scarvelis and Solomon [2023], our formulation allows us to output these geodesics, and does not require a separate training scheme; we elaborate on this point in the following subsection.

### 4.2 RELATED WORK

Although our setup is taken from Scarvelis and Solomon [2023], there are several differences between our work and theirs. They deploy a specialized duality theory based on section 2.1, where the Kantorovich potentials must be 1-Lipschitz with respect to the weighted Euclidean metric; this is enforced using an additional regularizer in the inner maximization problem in eq. (22). Finally to ensure the metric does not collapse, they add another regularizer on eq. (22), for the outer minimization problem. They use eq. (22) *only* to fit a metric $\hat{A}_\vartheta$, and later use another optimization problem (based on continuous normalizing flows) to fit their

geodesics. In contrast, our approach is self-contained, un-regularized, generalizable to other notions of cost, and we directly obtain approximations of the transport map with $y_\zeta$ and transport paths with $\varphi_\eta$.

## 5 CONCLUSION

In this work, we proposed an efficient framework for computing geodesics under generalized least-action principles, or Lagrangians, leading to large-scale computation of Lagrangian Optimal Transport trajectories. Combining amortization and the use of splines, we demonstrate the capacity of our method on a suite of problems, ranging from learning non-Euclidean geometries from data, to computing optimal transport maps under (known) non-Euclidean geometries and costs. There are many remaining fundamental research avenues that arise as a result of this work. Examples include a statistical analysis of these new costs (*e.g.*, Hundrieser et al. [2023], Hütter and Rigollet [2021]), extensions to the unbalanced optimal transport setting through the Wasserstein-Fisher-Rao metric [Gallouët and Monsaingeon, 2017], and extensions to multi-marginal optimal transport [Pass, 2015].

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

# Neural Optimal Transport with Lagrangian Costs
# (Supplementary Material)

**Aram-Alexandre Pooladian**[1,3]       **Carles Domingo-Enrich**[2,3]       **Ricky Tian Qi Chen**[3]       **Brandon Amos**[3]

[1]Center for Data Science, New York University
[2]Courant Institute of Mathematical Sciences, New York University
[3]FAIR, Meta

## A   DETAILS ON THE USE OF SPLINES FOR GEODESICS AND PATHS

This section provides more background information and details behind the amortized splines in section 3 that address challenge 2.

### A.1   CUBIC SPLINES

Cubic splines, *e.g.*, as reviewed in McKinley and Levine [1998], Wolberg [1988], Bartels et al. [1995], Weisstein [2008], Hastie et al. [2009], Burden et al. [2015], are a widely-used method for fitting a parametric function to data. We start with a review of general splines in one dimension (appendix A.1.1), then extend those to multiple dimensions (appendix A.1.2), then use those for representing the Lagrangian paths and geodesics (appendix A.2), then amortize those (appendix A.3).

### A.1.1   . . . in one dimension

In one dimension, a cubic spline is defined by

$$\gamma(x) = \begin{cases} \gamma_1(x) & \text{if } x_1 \leq x < x_2 \\ \gamma_2(x) & \text{if } x_2 \leq x < x_3 \\ \gamma_{n-1}(x) & \text{if } x_{n-1} \leq x < x_n \end{cases} \tag{29}$$

where $x \in \mathbb{R}$, $x_i$ for $i \in \{1, \ldots, n\}$ are the *knot points* and

$$\gamma_i(x) := a_i + b_i x + c_i x^2 + d_i x^3 \tag{30}$$

are the cubic component functions with coefficients $\bar{\varphi}_i := [a_i, b_i, c_i, d_i]$. We write the vector of all coefficients as $\bar{\varphi} := [\bar{\varphi}_1, \ldots, \bar{\varphi}_{n-1}]$.

**Challenge 3** (Parameterizing splines). *The coefficients $\bar{\varphi}$ are unknown and fit to data. While they could be taken directly as the parameters for $\gamma$, it would not result in a continuous function around the knot points.*

The standard approach to resolve these discontinuities is to constrain the component functions to be continuous and have matching values and derivatives

$$\gamma_i(x_{i+1}) = \gamma_{i+1}(x_{x+1}) \text{ for } i \in \{1, \ldots, n-1\}$$
$$\gamma_i'(x_{i+1}) = \gamma_{i+1}'(x_{x+1}) \text{ for } i \in \{1, \ldots, n-1\} \tag{31}$$
$$\gamma_i''(x_{i+1}) = \gamma_{i+1}''(x_{x+1}) \text{ for } i \in \{1, \ldots, n-1\}.$$

These constraints, along with other conditions can be used to provide a set of basis vectors $B := [b_i]_{i=1}^m$ where $b_i \in |\bar{\varphi}|$ of spline parameterizations $\bar{\varphi}$ that satisfy eq. (31), *e.g.*, as in Hastie et al. [2009, Section 5.2.1]. In other words, any linear combination of the basis vectors $b_i$ will result in a valid parameterization. We can thus reparameterize the spline with $\varphi \in \mathbb{R}^m$ to be based on linear combinations of the basis, providing

$$\bar{\varphi} = B\varphi = \sum b_i \varphi_i \tag{32}$$

The advantage of this reparameterization is that $\varphi$ is a parameterization of splines in the unconstrained reals and can therefore be treated as a standard learnable parameter for our geodesic computations.

### A.1.2 ...in multiple dimensions

The standard extension of splines to functions of multiple dimensions, *e.g.*, for graphics [Bartels et al., 1995], is to parameterize a one-dimensional spline eq. (29) on each coordinate. We will notate these as $\gamma_\varphi : \mathbb{R} \to \mathbb{R}^d$, $\gamma_\varphi(x) := [\gamma_{\varphi_1}(x), \ldots, \gamma_{\varphi_d}(x)]$ where $\varphi_i$ is the parameterization of the basis coefficients for each one-dimensional spline.

### A.2 CUBIC SPLINES FOR GEODESICS AND LAGRANGIAN PATHS

We follow Beik-Mohammadi et al. [2021], Detlefsen et al. [2021] and represent geodesics and Lagrangian paths between two points $x, y$ by a multi-dimensional spline $\gamma_\varphi(t)$ parameterized by $\varphi$ where $t \in [0, 1]$ is the time. The basis for the splines enforce the smoothness properties in eq. (31) as well as the boundary conditions $\gamma_\varphi(0) = x$ and $\gamma_\varphi(1) = y$.

### A.3 AMORTIZED CUBIC SPLINES FOR GEODESICS

Instead of computing the spline parameters $\varphi$ individually for every geodesic, we propose to *amortize* them across the geodesics needed for the OT maps. This results in parameterizing an amortization model $\varphi_\eta(x, y)$ that predicts the spline parameters for a geodesic between $x$ and $y$ that we learn with objective-based amortization in eq. (19).

## B  SYNTHETIC DATA FOR LAGRANGIANS WITH POTENTIALS

We consider five potential functions $U(x)$. The following four potential functions are from Koshizuka and Sato [2022]:

$$U_{\text{box}}(x) := -M_1 \cdot \mathbf{1}_{[-0.5,0.5]^2}(x), \tag{33}$$

$$U_{\text{slit}}(x) := -M_2 \cdot (\mathbf{1}_{([-0.1,0.1],(-\infty,-0.25))}(x_1, x_2) + \mathbf{1}_{([-0.1,0.1],[0.25,\infty))}(x_1, x_2)), \tag{34}$$

$$U_{\text{hill}}(x) := -M_3 \|x\|^2, \tag{35}$$

$$U_{\text{well}}(x) := -M_4 \exp(-\|x\|^2), \tag{36}$$

where $M_1, M_2, M_3$ and $M_4$ are constants.

The Gaussian-mixture example is taken from Liu et al. [2022], which amounts to the following potential function

$$U_{\text{GMM}}(x) := -M_5 \sum_{i=1}^3 \mathbf{1}_{B_i}(x), \tag{37}$$

where $B_i := \{x : \|x - m_i\| \le 1.5\}$ with $m_i \in \{(6, 6), (6, -6), (-6, -6)\}$.

We approximate the hard constraints using sigmoid functions. We make the choices $M_1 = 0.01$, $M_2 = 1$, $M_3 = 0.05$, $M_4 = 0.01$, $M_5 = 0.1$ — we are unable to use the same choice of $M$ for all potentials as a result of numerical instabilities that arise in the geodesic computation.

## C  DATA FROM Scarvelis and Solomon [2023]

We briefly outline the three datasets used in section 4, all of which were taken directly from Scarvelis and Solomon [2023], following their open source repository https://github.com/cscarv/riemannian-metric-learning-ot; here we simply explain the data generating processes.

The three datasets have a similar flavor: Let $\gamma$ be a time-varying curve, and suppose we have access to the matrix function $A(\cdot)$ which generates the known geometry. This allows the authors to generate a velocity field between two fixed points $x$ and $y$ (respectively, initial and final position of $\gamma$) using the following optimization problem

$$\min_{\theta} \int_0^1 \|v_{(t,\theta)}(\gamma_t)\|^2_{A(\gamma_t)} \, \mathrm{d}t + \|\gamma(1) - y\|, \tag{38}$$

where $v_{(t,\theta)}(\cdot)$ is a time-varying neural network (parametrized by $\theta$) that is the solution to a neural ODE, where they also enforce the initial condition $\gamma_0 = x$. The integral in time is replaced with a sum over indices $0 = t_1 < t_2 < \ldots < t_m = 1$. For a given collection of samples from measures $\{\rho_i\}_{i=1}^{K-1}$, the authors randomly pair up the data and solve eq. (38) across batches using the Chen [2018] package (specifically using `odeint`). Equation (38) is solved using AdamW with a learning rate of $10^{-3}$ and weight-decay factor $10^{-3}$, with 100 epochs of training per pair of samples. The learned solution $v_{(t,\hat{\theta})}$ is able to generate data at various time-points. With this setup in mind, we can turn to precise details for the three datasets.

**Circular trajectory**  The circular path is enforced using the matrix

$$A(x) = \begin{pmatrix} \frac{x_1^2}{\|x\|^2} & 1 - \frac{x_1 x_2}{\|x\|^2} \\ 1 - \frac{x_1 x_2}{\|x\|^2} & \frac{x_2^2}{\|x\|^2} \end{pmatrix}. \tag{39}$$

The goal is to generate Gaussian data that flows according to $A$. To this end, the authors fix four possible means (in order) $\mu \in \{(1,0), (0,1), (-1,0), (0,-1)\}$, and fix $\sigma := 0.1$, which define $\rho_i := N(\mu_i, \sigma^2)$. 100 samples are drawn from each $\rho_i$, which constitutes the finite-samples that are used in the objective function eq. (38). Once the velocity field is learned, there are 24 equispaced time-points from which they draw samples, resulting in 24 Gaussian distributions that flow according to $A$.

**Mass-splitting trajectory**  In this example, $A(x) = I - w(x)w^\top(x)$, with

$$w(x) = \begin{cases} \left(\frac{1}{\sqrt{2}}, \frac{1}{\sqrt{2}}\right) & x_2 \geq 0, \\ \left(\frac{1}{\sqrt{2}}, \frac{-1}{\sqrt{2}}\right) & x_2 < 0. \end{cases} \tag{40}$$

In this case, there are three Gaussians, with means $\mu_i \in \{(0,0), (10,10), (10,-10)\}$ and unit variance. Again, 100 samples are drawn from each, which are randomly paired and allow the authors to numerically solve eq. (38). Once they have a learned vector field, they generate the data at 10 equispaced time-points.

**X-path trajectory**  In this third case example, $A(x) = I - w(x)w^\top(x)$, with

$$w(x) = \alpha(x)w_1(x) + \beta(x)w_2(x) \tag{41}$$

where $w_1(x) = \left(\frac{1}{\sqrt{2}}, \frac{1}{\sqrt{2}}\right)$ and $w_2(x) = \left(\frac{1}{\sqrt{2}}, \frac{-1}{\sqrt{2}}\right)$, and $\alpha(x) = 1.25 \tanh(\mathrm{ReLU}(x_1 x_2))$ and $\beta(x) = -1.25 \tanh(\mathrm{ReLU}(-x_1 x_2))$. Here, there are two sets of two trajectories, corresponding to Gaussian data with means $\mu_i^{(1)} \in \{(-1,-1), (1,1)\}$ and $\mu_i^{(2)} \in \{(-1,1), (1,-1)\}$, all with standard deviation $\sigma = 0.1$. As before, 100 samples are generated, and eq. (38) is solved (twice) numerically; 10 time-points per velocity field are used to generate the total data.

# D HYPER-PARAMETERS

Table 3: Hyper-parameters for computing the OT maps in figs. 1 and 2 with alg. 1.

| Hyper-Parameter | Value |
|---|---|
| Number of spline knots | 30 |
| $g_\theta$ MLP layer sizes | `[64, 64, 64, 64]` |
| $y_\zeta$ MLP layer sizes | `[64, 64, 64, 64]` |
| $\gamma_\varphi$ MLP layer sizes | `[1024, 1024]` |
| MLP activations | Leaky ReLU |
| $g_\theta$ learning rate schedule | Cosine (starting at $10^{-4}$ and annealing to $10^{-2}$) |
| $y_\zeta$ learning rate schedule | Cosine (starting at $10^{-4}$ and annealing to $10^{-2}$) |
| $\gamma_\varphi$ learning rate | $10^{-4}$ (no schedule) |
| Batch size | 1024 |
| $c$-transform solver | LBFGS (20 iterations, backtracking Armijo line search) |

Table 4: Hyper-parameters for computing figs. 3 and 4 and table 2 with alg. 1 and eq. (22).

| Hyper-Parameter | Value |
|---|---|
| Number of spline knots | 30 |
| $g_\theta$ MLP layer sizes | `[64, 64, 64, 64]` |
| $y_\zeta$ MLP layer sizes | `[64, 64, 64, 64]` |
| $\gamma_\varphi$ MLP layer sizes | `[1024, 1024]` |
| MLP activations | Leaky ReLU |
| $g_\theta, y_\zeta, \gamma_\varphi$ learning rates | $10^{-4}$ no schedule |
| Batch size | 1024 |
| $c$-transform solver | LBFGS (20 iterations, backtracking Armijo line search) |
| $A_\vartheta$ learning rate | $5 \cdot 10^{-3}$ |
| Update frequency | 1 update of $A_\vartheta$ for every 10 updates of $g_\theta, y_\zeta$, and $\gamma_\varphi$ |