# OpenReview forum: "Neural Optimal Transport with Lagrangian Costs"
_auai.org/UAI/2024/Conference — UAI 2024 poster_

### Official Review · Reviewer_M6tz · 2024-03-21

**Q2-1 Originality-Novelty:** 3
**Q2-2 Correctness-Technical Quality:** 4
**Q2-5 Clarity Of Writing:** 4

**Q1 Summary And Contributions:**

- This paper proposes a framework for performing dynamical optimal transport with Lagrangian-based costs, a significantly more general setting than the standard Euclidean cost setting.
- The OT problem is phrased via Kantorovich potentials, but naively solving this dual problem is computationally intractable due to the presence of a nested optimization problem (i.e., an optimization problem is required just to evaluate the cost between two points)
- Accordingly, the authors propose a method largely based on amortized optimization in order to tractably solve this problem.
- Some simple experiments on 2D datasets with a given potential or Riemannian metric are presented. The authors also perform some experiments on a 2D metric learning task.

**Q2-3 Extent To Which Claims Are Supported By Evidence:**

3: Good: the main claims are supported by convincing evidence (in the form of adequate experimental evaluation, proofs, (pseudo-)code, references, assumptions).

**Q2-4 Reproducibility:**

3: Good: key resources (e.g. proofs, code, data) are available and key details (e.g. proofs, experimental setup) are sufficiently well-described for competent researchers to confidently reproduce the main results.

**Q3 Main Strengths:**

- The paper solves an interesting and potentially impactful problem, namely OT with Lagrangian-based costs. There are many problems where an L2 cost is an overly simplistic assumption, and this work is a step forward in the applicability of OT methods to real-world problems. Most other work in this area focuses on the L2 cost, and this generalization is likely to be useful.
- The paper is technically sound, with the necessary conditions for existence of solutions to the OT problem clearly specified.
- The experiments clearly demonstrate the methodology  and are compelling.
- Throughout, the writing and exposition are atypically clear. The examples in 2.2 are greatly appreciated.

**Q4 Main Weakness:**

- The techniques themselves are relatively minor extensions of existing methods, particularly the amortized optimization techniques of [Amos 2023](https://arxiv.org/abs/2210.12153).
- The weakest point of the paper is a lack of real-world experiments. Although the experiments serve as a nice demonstration of the technique, they are limited to a fairly simple 2D setting. This leaves me wondering if it is difficult to scale this method up to higher dimensional datasets.
- Along similar lines, a more extensive discussion of how difficult, empirically, it is to train the method is lacking. That is, the method involves several nested optimization problems, which seems challenging -- is the method sensitive to hyperparameters / neural network architecture choices? How much data is needed to reliably fit these models? How important is the fine-tuning step when solving for the $c$-potentials?
- The paper focuses on only two possible types of Lagrangians (i.e., induced by potentials or by a non-Euclidean metric). However, the framework seems to work for more general Lagrangians.

**Q5 Detailed Comments To The Authors:**

- One thing I found unclear was that, in Section 2.1, the assumptions appear to only guarantee the existence of an optimal *coupling*, not an optimal map. However, such optimal maps are discussed after Equation 6, and used throughout the paper. Could the authors please clarify this, e.g. is it implicitly assumed that there indeed exists an optimal map?
- It was a little unclear to me why Equation 15 is valid (i.e. interchanging the gradient and integrals) -- further justification of this step would be appreciated.
- In Equation 26, I do not see where the motivation for this specific form of $B$ comes from. An experiment which explores other parametrizations of Equation 26 could be useful; e.g. what would happen if you instead parametrize the network to predict an arbitrary matrix, followed by projection onto the manifold of SPD matrices?


Minor points / Typos:
- Equation 17 should probably be $\min_\zeta$
- Equation 8 seems to have an extra pair of curly brackets since the RHS is a single scalar value
- After Equation 27, "While this seems quite ad hoc appears limited to rotations”
- Minor error in Figure 2 in the box "contours indicate..."


Related work:
- I highly recommend the authors compare their approach (at least including a discussion in the related work) to [Neklyudov et al., 2023](https://arxiv.org/abs/2310.10649).

**Q9 Complying With Reviewing Instructions:**

Yes

---

> ### Author Rebuttal · Authors · 2024-04-04
>
> Thank you for the incredibly detailed and kind review!
>
> First, all typos will be addressed and a lengthier discussion regarding the paper by Neklyudov et al. will be included as well; at the time of writing, their paper was not released but we agree that a comparison should be had, the omission was largely out of space constraints. Their method does not result in a deterministic mapping for these cost functions, which is the goal of our article.
>
>
> *Experimental details:*
> We will include a more extensive discussion of the experimental details that were originally left out of the paper; this was done in order to help convey intuition. Indeed, the method concurrently solves several nested optimization problems:
> - We found most of the hyper-parameters and architectural choices to be consistent with prior work on static neural optimal transport solvers, which makes sense to us as ours is an extension from the squared-Euclidean setting. In the squared-Euclidean OT setting, there is an architectural choice of parameterizing the potentials with an input-convex neural network, but in Lagrangian and general cost settings it’s unclear how to parameterize a general and expressive $c$-convex function.
> - We also found the data requirements of the potentials to be consistent with Neural Euclidean OT approaches. The potentials can roughly be fit to ~1000s  samples from the measures and require ~10-20k parameter updates for training.
> - The fine-tuning step of the $c$-transform is important as without it the potential can over-optimize the dual objective. We found this to be consistent with fine-tuning in the squared-Euclidean setting. By the end of training, the $c$-transform amortizer reasonably predicts the transform as in e.g.,Figure 3 of [Amos, ICLR 2023]. But during training, the prediction is worse and the fine-tuning step substantially helps correct it.
>
>
> *On the detailed comments:*
> - The discussions in Section 2.1 guarantee the existence of a deterministic coupling, which results in a deterministic mapping. Moreover, the optimal transport map is that given by (6).
> - Exchanging differentiation and integration is justified since the function of interest is differentiable with respect to the model parameters and is independent of the integration. If one wants to be completely rigorous, this can be carefully justified using dominated convergence theorem, but this is typically omitted from discussions in the machine learning community.
> - For the metric learning experiments, our choice of the rotational metric (with $B$) comes from us parameterizing a metric where one direction takes a higher value (i.e., 10x) than the direction orthogonal to it. Then, given this constraint, the metric learning is able to rotate the metric so the trajectories from the real data coincide with the lower directions of this skewed metric. We tried more general parameterizations of the metric here with a neural network, but found it to be unstable. This is because our method requires near-exact geodesic computations, which is in general an extremely challenging problem itself for general neural network (or otherwise complicated) metrics.

---

### Official Review · Reviewer_zDck · 2024-03-22

**Q2-1 Originality-Novelty:** 2
**Q2-2 Correctness-Technical Quality:** 2
**Q2-5 Clarity Of Writing:** 3

**Q1 Summary And Contributions:**

This work considers the optimal transport problem between probability measures when the underlying cost function satisfies the least action principle – Lagrangian cost. These notions are useful when connecting observations from a physical system, where the transport dynamics are influenced by the geometry of the systems, such as obstacles. The paper aims to provide a computational framework that allows practitioners to enforce transport with a more general cost that can incorporate such geometries.

**Q2-3 Extent To Which Claims Are Supported By Evidence:**

2: Fair: the main claims are somewhat supported by evidence (but the experimental evaluation may be weak, or does not match entirely with the claims, important baselines may be missing, proofs contain important ideas but lack rigor, algorithmic details are only discussed superficially, references are imprecise, assumptions are not sufficiently motivated or explicated, etc.).

**Q2-4 Reproducibility:**

2: Fair: key resources (e.g. proofs, code, data) are unavailable but key details (e.g. proof sketches, experimental setup) are sufficiently well-described for an expert to confidently reproduce the main results.

**Q3 Main Strengths:**

The paper considers an important problem in computing the optimal transport between the probability measure. It gives a numerical solution when the underlying cost of displacement is governed by a least action principle. Their contributions are two-fold:

(1) Compute the Lagrangian optimal transport maps,

(2) Compute the resulting paths for these maps.

**Q4 Main Weakness:**

The setting of the paper is taken from Christopher Scarvelis and Justin Solomon, ICLR, 2023 and empirically shows comparison with their work. Probably experiments on more datasets such as eBird basic dataset, snow goose datasets, could further strengthen their claim. Also, the paper numerically compares with the work of Scarvelis and Solomon, 2023. Including some more baseline, such as the following paper, could again make the experimental part more comprehensive.

Optimal transport analysis of single-cell gene expression identifies developmental trajectories in reprogramming. Schiebinger et, al, 2019.
Also, there are some recent related works with which the author can consider comparing their work, such as “Neural Optimal Transport, ICLR, 2023.”

**Q5 Detailed Comments To The Authors:**

Pls see in response to Q4.

**Q9 Complying With Reviewing Instructions:**

Yes

---

> ### Author Rebuttal · Authors · 2024-04-04
>
> Thank you for your review.
>
> The difficulty in our work is the modeling of the cost function, which requires solving an optimization problem per pair of datum (x,y). For any other method that we were aware of (including the two papers you mentioned in your review), the cost function needs to be deterministic, which drastically changes how the optimization is performed.
>
> For example, the paper by Schiebinger et al. uses (entropic) optimal transport couplings between consecutive time points. The goal of this section of the paper was metric learning, which their approach does not allow for; they require the metric to be known a priori.
>
> With regards to the paper by Korotin and co-authors, the two methods are actually very similar. Again, the main difference lies in the cost function: theirs is deterministic, and ours requires another minimization problem. There are several distinctions on a computational technical level that are different (ours is more closed based on [Amos, ICLR 2023]), but at a high-level, these are the same approach and a comparison does not make sense in the context of our work.

---

### Official Review · Reviewer_fDbx · 2024-03-24

**Q2-1 Originality-Novelty:** 2
**Q2-2 Correctness-Technical Quality:** 3
**Q2-5 Clarity Of Writing:** 3

**Q10 Ethical Concerns:**

No concerns.

**Q1 Summary And Contributions:**

The paper presents and algorithm for so-called Lagrangian optimal transport, where the cost is given implicitly from the minimal action principle. The goal is to compute optimal transport map and their paths (that solve the infimum problem as I understand).

The work proposes an algorithm for this task and tests it on several model examples.

**Q2-3 Extent To Which Claims Are Supported By Evidence:**

2: Fair: the main claims are somewhat supported by evidence (but the experimental evaluation may be weak, or does not match entirely with the claims, important baselines may be missing, proofs contain important ideas but lack rigor, algorithmic details are only discussed superficially, references are imprecise, assumptions are not sufficiently motivated or explicated, etc.).

**Q2-4 Reproducibility:**

3: Good: key resources (e.g. proofs, code, data) are available and key details (e.g. proofs, experimental setup) are sufficiently well-described for competent researchers to confidently reproduce the main results.

**Q3 Main Strengths:**

The problem that is studied is interesting for several optimal transport applications, i.e., on manifolds.
The main application seems to be metric recovery (Table 2), which shows improvement over prior work.

**Q4 Main Weakness:**

The numerical experiments look rather weak in my opinion, i.e. Table 1 does not show any improvement (although some lines are highlighted, it does not seem to be better, especially if you take standard deviation into account).

The algorithm seems to be a rather straightforward adaptation of existing approaches.

**Q5 Detailed Comments To The Authors:**

1. Why 'filling the gap' (main motivation) is of practical interest? As far as I understand, it is for the metric learning?
2. What are the competitive approaches?
3. The numerical experiments are quite model, what are the limitations and/or large scale application of you models?

**Q9 Complying With Reviewing Instructions:**

Yes

---

> ### Author Rebuttal · Authors · 2024-04-04
>
> Thank you for the review.
>
> Our primary contribution compared to other approaches is the learning of transport maps for these cost functions. When learned, maps are cheaper to deploy as they only require a single function evaluation to reach the target, whereas methods based on flows require significantly more model evaluations. At the time of writing this article, there were no other deterministic approaches for learning optimal transport maps for Lagrangian costs, and so there were no other competitive comparisons. While metric learning is one application of this, there are many other physically-motivated settings where it could be interesting to perform optimal transport in non-Euclidean spaces for navigating particle systems with constraints. We hope our computational methods will help further advance this nascent field.
>
> We agree that the numerics are not jaw-dropping, but we respectfully disagree with your comment indicating that our method shows “any improvement” in Table 1. You are correct that the “hill” example was quite difficult, but apart from this, our method does outperform the others, even when accounting for standard deviation. Finally, it is worth pointing out that the fairest comparison would be with the “NLSB (expected)” row, as it is also a deterministic mapping. Though, their deterministic map is an average of several trajectories, with each trajectory requiring 100 steps to produce a transport path/map. Against this approach, our method is significantly better with respect to our considered metrics.
>
> While the metric-learning tasks (Table 2) are small-scale and synthetic, they are the same as other accepted, peer-reviewed publications.  We are not aware of other settings that perform metric learning via OT where there is a known ground-truth metric to compare against. We believe understanding and measuring the performance of metric learning approaches on these simpler settings are important for us as a community to gain intuitions on the modeling choices as we continue developing and scaling up into more complex applications.

---

### Official Review · Reviewer_WmuN · 2024-03-24

**Q2-1 Originality-Novelty:** 2
**Q2-2 Correctness-Technical Quality:** 3
**Q2-5 Clarity Of Writing:** 3

**Q1 Summary And Contributions:**

This paper studies the optimal transport problem when the cost is based on a least action principle, known as Lagrangian cost. This approach is relevant for physical systems where transport dynamics are shaped by system geometry, such as obstacles, enabling the inclusion of a priori system knowledge. The paper presents a contribution by solving these optimal transport problems without regularization, showcasing efficient computation of geodesics and spline-based paths. The effectiveness of this method is validated through synthetic examples from the literature.

**Q2-3 Extent To Which Claims Are Supported By Evidence:**

3: Good: the main claims are supported by convincing evidence (in the form of adequate experimental evaluation, proofs, (pseudo-)code, references, assumptions).

**Q2-4 Reproducibility:**

2: Fair: key resources (e.g. proofs, code, data) are unavailable but key details (e.g. proof sketches, experimental setup) are sufficiently well-described for an expert to confidently reproduce the main results.

**Q3 Main Strengths:**

1. The paper studies optimal transport with Lagrangian Costs, which is important problem recently. Recent works rarely provide
deterministic mappings and optimal paths.  And this works fills this gap in the literature on computational optimal transport.
2. I am not an expert on optimal transport for dynamic systems. The paper structure and mathematical formulation look reasonable to me.

**Q4 Main Weakness:**

1. In the paper, it is less clear when the situation occurs in practice. Can authors also provide an example from a AI/ML application perspective, i.e., cellular dynamics , time series modeling or diffusion models?
2. Codes is not available.

**Q5 Detailed Comments To The Authors:**

See above

**Q9 Complying With Reviewing Instructions:**

Yes

---

> ### Author Rebuttal · Authors · 2024-04-04
>
> Thank you for your kind review! We apologize that our code is not currently publicly available, but it will be made available upon acceptance.
>
> With regards to practical considerations: the cost function is often a modeling choice, and Lagrangian costs represent a general parameterization arising in physical and geometric systems. For example, for cellular dynamics, most works have only considered the squared-Euclidean cost of displacement, and applied many techniques from optimal transport as a result. Though, with more domain knowledge, it would be possible to construct potential functions (read as $U$ in our paper; see for example https://arxiv.org/abs/2204.04853 or https://arxiv.org/abs/2310.10649) that deter or encourage motion in certain regions. The problem setting also connects to recent ML and diffusion modeling trends on doing generative modeling on manifolds (see e.g., https://openreview.net/forum?id=g7ohDlTITL https://proceedings.neurips.cc/paper_files/paper/2023/hash/fe1ab2f77a9a0f224839cc9f1034a908-Abstract-Conference.html ), as well as constrained settings. Our paper shows how to solve static OT problems between measures in these non-Euclidean settings.

---

### Meta-Review · Area_Chair_DVHV · 2024-04-15

Summary: The paper investigates the optimal transport problem when the cost is derived from a least action principle, known as the Lagrangian cost. This approach is particularly relevant for physical systems where the transport dynamics are influenced by system geometry, such as the presence of obstacles, allowing for the incorporation of a priori knowledge about the system. The paper's main contribution lies in solving these optimal transport problems without regularization, demonstrating efficient computation of geodesics and spline-based paths. The effectiveness of this method is validated through synthetic examples drawn from the existing literature.

Meta-review: Based on the initial reviews, the verdict is that the authors have crafted an above-the-bar paper that presents a compelling proposal backed by solid reasoning. There is one pending borderline reject, but I believe the authors have handled these issues sufficiently. Based on the interaction between authors and reviewers, most concerns have been raised, and all the reviewers place the paper above the acceptance borderline. The authors have adequately answered and interacted with the authors' questions.